# Individual-, household- and community-level determinants of infant mortality in Ethiopia

**Girmay Tsegay Kiross**  [1,2]*, **Catherine Chojenta**[2], **Daniel Barker**[3], **Deborah Loxton**[2]

**1** Department of Public Health, College of Health Sciences, Debre Markos University, Debre Markos, Ethiopia, **2** Research Centre for Generational Health and Ageing, Faculty of Health and Medicine, University of Newcastle, Newcastle, New South Wales, Australia, **3** School of Medicine and Public Health, Faculty of Health and Medicine, University of Newcastle, Newcastle, New South Wales, Australia

* girmshe@gmail.com, Girmay.kiross@uon.edu.au

## Abstract

### Introduction

People living in the same area share similar determinants of infant mortality, such as access to healthcare. The community's prevailing norms and attitudes about health behaviours could also influence the health care decisions made by individuals. In diversified communities like Ethiopia, differences in child health outcomes might not be due to variation in individual and family characteristics alone, but also due to differences in the socioeconomic characteristics of the community where the child lives. While individual level characteristics have been examined to some extent, almost all studies into infant mortality conducted in Ethiopia have failed to consider the impact of community-level characteristics. Therefore, this study aims to identify individual and community level determinants of infant mortality in Ethiopia.

### Method

Data from the Ethiopian Demographic and Health Survey in 2016 were used for this study. A total of 10641 live births were included in this analysis. A multi-level logistic regression analysis was used to examine both individual and community level determinants while accounting for the hierarchal structure of the data.

### Results

Individual-level characteristics such as infant sex have a statistically significant association with infant mortality. The odds of infant death before one year was 50% higher for males than females (AOR = 1.66; 95% CI: 1.25–2.20; p-value <0.001). At the community level, infants from pastoralist areas (Somali and Afar regions) were 1.4 more likely die compared with infants living in the Agrarian area such as Amhara, Tigray, and Oromia regions; AOR = 1.44; 95% CI; 1.02–2.06; p-value = 0.039).

### Conclusion

Individual, household and community level characteristics have a statistically significant association with infant mortality. In addition to the individual based interventions already in

**Data Availability Statement:** We used data from Ethiopian Demographic and Health survey. The data are available and can be downloaded from the Measure DHS website www.measuredhs.com.

**Funding:** The authors received no specific funding for this work.

**Competing interests:** The authors have no competing interests.

place, household and community-based interventions such as focusing on socially and economically disadvantaged regions in Ethiopia could help to reduce infant mortality.

## Introduction

The IMR is an essential national indicator of health because it is very sensitive to general structural elements, particularly to socio-economic development and basic living conditions [1]. It is also an essential measure of women's and children's wellbeing [2]. The IMR is a proxy measure of population health [3]. It can also indicate the health status of women, the quality of and access to healthcare services, public health practices and socio-economic conditions in any given population [4]. A high IMR may also indicate lack of proper care for children due to poverty, lack of parents' education and societal preferences (such as the preference for a male child) [5]. In Ethiopia, infant mortality is an important public health problem, where 48 per 1000 live births have resulted in death during the first year of life in 2016 [6]. To attain SDG 3, infant mortality has to reduce substantially, because it accounts for 75% of all under-5 mortality [7].

In Ethiopia, a number of recent studies have been conducted on the determinants of infant mortality. For example, a matched case–control study in south-west Ethiopia in 2012, focusing on a handful of communities (mostly small rural districts), found that antenatal care (ANC) follow-up, handwashing with soap before feeding children, birth size, the mother's perception of modern medical treatments, birth order and preceding birth interval were determinants of infant mortality [8]. Another research project conducted at the institutional level in northern Ethiopia showed that the death of an infant in early age was associated with low birth weight, low breastfeeding practice and maternal complications [9]. However, the study ignored the impact of socio-economic determinants in relation to infant mortality, which limits the applicability of the results on a large scale. Furthermore, a number of recent studies have investigated the determinants of infant mortality at the national level. For example, national studies conducted using EDHS data in 2000, 2005 and 2016 found that the mother's age, mother's level of education, child birth order, source of drinking water and sex of the infant had statistically significant associations with infant mortality [10, 11]. However, several of these studies have assumed the independence of individual-, household- and community-level factors, and researchers have not considered the clustered nature of EDHS data.

The sampling procedure in the EDHS is multistage cluster sampling, by which individuals were nested in clusters and infant mortality be correlated with these clusters [6]. This violates the assumption of independence, which could introduce a serious bias in programmatic implementation, implying that contextual variables are not considered in the study. Contextual variables such as the classification of respondents' residence (as urban or rural) allow researchers to study how a wide range of surrounding characteristics may influence health and wellbeing. For example, researchers have found that geographical access to health care has an effect on infant mortality [12].

An analytical study in developing countries has indicated that infant mortality is affected by several community-level determinants, such as the community's level of education, geographical isolation, poverty rate, community access to skilled maternal health services, area of residence and the presence of community media [13]. Evidence also suggests that the community in which a child is born determines their survival status [14–16]. In a synthesis of health survey data from 28 countries, it was shown that being a member of specific families and

communities determined the child's survival in SSA countries, implying that the socio-economic characteristics of the community in which the infant resides are important determinants of infant mortality [17, 18]. In a national study in Malawi, it was found that there was variation in child mortality at the household and community levels; it was shown that infant mortality varied by 18% when community-level characteristics were added to individual-level characteristics [19]. In a nationally representative cross-sectional study of 28,647 live births in Nigeria also showed that 16.7% of the variance in the risks of infant mortality across community level characteristics [20].

While several studies on infant mortality have been conducted in Ethiopia, many of these have given more attention to the influence of individual-level attributes (characteristics of infants and mothers) and less to the community-level determinants of infant mortality [21–23]. Individuals within a community share similar resources due to most public services being spatially organised into clusters; this may determine the health of individuals who live in that community [17]. It is widely acknowledged that differences in child health outcomes might not be due to variation only in family characteristics, but in the socio-economic characteristics of the community in which the child lives [24, 25]. Infant mortality might vary at the community level due to variations in access to the nearest health and educational institutions, means of communication and the number of households with electricity, piped water and sewers [26]. People living in the same area share some of the major determinants of infant mortality, such as access to water, sanitation and health care. A community's prevailing norms and attitudes about health behaviours can also influence the healthcare decisions made by individuals [27].

Most of the studies conducted in Ethiopia have not considered community-level determinants of infant mortality [8, 10, 21]. However, studies conducted in diversified communities such as Ethiopia should take into account the variation at the community level to address the high rate of infant mortality in the nation. That is, individual-, household- and community-level factors need to be considered to ascertain the relative importance of interventions on child health. This study aims to identify individual-, household- and community-level determinants of infant mortality in Ethiopia using data from the 2016 EDHS. Based on previous research, it is expected that both individual- and community-level characteristics will be significantly associated with infant mortality.

## Materials and methods

### Study area and setting

The source of data for this study was the 2016 EDHS. The data were downloaded from the MEASURE DHS website (www.meauredhs.com). The survey covered all nine regions and two city administrations of Ethiopia.

### Study design and sampling

The survey design of the 2016 EDHS was cross-sectional, and survey participants were selected through a stratified two-stage cluster sampling technique. The full details of the methods and procedures of data collection in the EDHS have been published elsewhere [28]. The survey collected information from a nationally representative sample of 16,650 households, including 15,683 women aged 15–49 years. The study population for the present study was 10,641 children who had been born in the 5 years preceding the survey, nested within 645 communities across the country. EDHS data collection took place from 18 January 2016 to 27 June 2016.

## Study variables

**Outcome variable.** The outcome variable for this study was whether a child had died before celebrating their first birthday by the time of interview with their parent.

**Explanatory variables.** These are the characteristics of a community or cluster. A community comprises of people living in a particular area or in a common location. In the 2016 Ethiopian Demographic and health survey programmes, the primary sampling units (PSU) are considered as proxies for communities or clusters [6]. **(See Table 1).**

## Data analysis

**Descriptive analysis.** Sampling weights were applied to adjust for the disproportionate allocation of the sample to the nine regions and two city administrations as well as the sample difference between urban and rural areas [6]. Descriptive statistics such as frequencies and proportions were calculated after applying these sampling weights.

**Multivariable multilevel analysis.** The EDHS used a multistage cluster sampling technique, whereby data were hierarchical (i.e., mothers and infants were nested within households, and households were nested within clusters). Considering the hierarchical nature of EDHS data, mothers and infants who lived within the same cluster may have had similar characteristics to other mothers and infants compared to those in other parts of the country. Considering the clustered sampling approach, a two-stage multivariable multilevel logistic regression analysis was used to estimate the effects of individual-household- and community-level determinants on infant mortality. Backward stepwise multilevel logistic regression analysis was performed to select individual-, household- and community-level variables to each model and those variables with *p-value* > 0 .25 were removed.

The fixed effect sizes of individual-, household- and community-level determinants on infant mortality were expressed as AORs with 95% confidence intervals. A *p* value of .05 was used as the cut-off for statistical significance. Additionally, the measure of variance (random effects) was reported in terms of the intraclass correlation coefficient [35] and proportional change in variance [36].

## Ethical considerations

Ethical clearance was obtained from the Human Research Ethics Committee of the University of Newcastle (Reference no.: H-2018-0386), and an approval letter for the use of the EDHS data set was gained from MEASURE DHS. No information obtained from the data set was disclosed to any third party.

# Results

## Characteristics of the study population

The general characteristics of the study population are shown in Tables 2–4. Approximately 11,023 women who had given birth to a child in the 5 years preceding the survey in Ethiopia, living in 643 different communities, were interviewed to obtain information on under-5 mortality. More than half of the women interviewed were aged 25–34 years, only 7.1% had attended secondary education or above, and more than half had not been employed in their lifetime. The majority of their partners were engaged in agricultural activities for employment, and only 12% of the men attended secondary education or above. A quarter of the women were first married at an aged less than 15 years old, and 89% were living in a rural area. The majority of the women gave birth to the index child when they were aged 20–34 years;

**Table 1. Description and measurement of individual-, household- and community-level exposure variables.**

| Variable | Category/Measurement/Definition |
|---|---|
| **Individual-level** | |
| Infant sex | Categorised as 'male' or 'female'. |
| Birth order | Mothers and caregivers were asked about all of their children's dates of birth. These were re-coded as 'first birth order', 'second and third birth order', 'fourth to sixth birth order' and 'seventh and above birth order' [28, 29]. |
| Preceding birth interval | The difference in months between the current birth and the previous birth, counting twins as one birth, based on the self-reported dates of birth from the mother or caregiver. The length of the preceding birth interval was calculated as the difference in the century day code for the date of birth for the index child and the preceding child, in days, then divided by 30.4375 to convert to months [28, 29]. Birth interval was categorised as (i) 'less than 24 months' (or short birth interval), (ii) 'between 24 and 59 months' (or recommended birth interval) and (iii) 'greater than 60 months' [28, 29]. The century day code is analogous to the century month code and gives the number of days since the beginning of 1900. A century month code (CMC) is the number of the month since the start of the century. For example, January 1900 is CMC 1, January 1901 is CMC 13, and January 1980 is CMC 961 [30]. |
| Reported birth size | Measuring the actual weight of the newborn child was very challenging because only one-fourth of Ethiopian mothers gave birth at health facilities; the remaining three-fourth of Ethiopian mothers gave birth at home. In the survey, mothers asked to estimate the size of their child based on their experience or by comparison with a previous child. The response options 'greater than average', 'average', 'smaller than average' and 'very small'. This variable was re-coded as 'above average', 'average' and 'below average' (smaller than average or very small) [28, 29]. |
| Multiple births | The original question asked about the number of births for one pregnancy, and the response option was open. This was re-coded as 'single birth' for one birth or 'multiple birth' for more than one birth [28, 29]. |
| Total live births | The original question asked for the number of live births in the mother's lifetime, and the response option was the number of living children based on the mother's self-report. This was re-coded based on quartiles as (i) 'one live birth', (ii) 'two to three live births', (iii) 'four to five live births' and (iv) 'more than five live births'. If the respondent was pregnant during the interview, one was added to the total live births [28, 29]. |
| Age of mother | The self-reported age of the mother (in years) when completing the survey. The reproductive age of mothers was categorised into seven categories as '15–19', '20–24', '25–29', '30–34', '35–39', '40–44' and '45+'. For this study, the reproductive age of the mother was re-coded as 'young' (15–24 years), 'young adult' (25–34 years) and 'middle age' (35–49 years) to obtain adequate samples for each category [28, 29]. |
| Marital status | The original question asked for the marital status of the respondents, and the response options were 'never married', 'married', 'living together', 'separated' and 'widowed' [31]. In this study, 'never married persons' are persons who never got married in concordance with valid regulations. "Married persons" are those who got married before a competent body in concordance with valid regulations. "Widowed persons" are persons whose marriage ceased to exist by death of one of spouses or by declaring a missing spouse dead respectfully. "Divorced persons" are those whose marriage was terminated. "Separated persons" had previously lived with a partner but were not currently living with a partner. 'living together persons' are those who living together but have not valid regulations [28, 29]. |
| Religion | Participants were asked to report their religion as either 'Orthodox', 'Catholic', 'Protestant', 'Muslim' or 'Traditional or other'. Orthodox, Catholic and Protestant were re-categorised as 'Christian'; the categories of 'Muslim' and 'Traditional or other' were retained [28, 29]. |
| Women's education level | The original question asked women about the highest level of education they had attended, and the response options were 'no education', 'primary education', 'secondary education' and 'higher education'. We re-coded these into 'no education', 'primary education' and 'secondary education and above'; we merged secondary and higher education because few had reported having attended higher education [28, 29]. |

*(Continued)*

**Table 1.** (Continued)

| Variable | Category/Measurement/Definition |
|---|---|
| Decision-making autonomy | Participants were asked about who usually decides to obtain healthcare, to purchase household items and to visit relatives. The response options were 'respondent' or 'partner'. The new variable combined scores for the three items. The value of each characteristic was 0 (partner) or 1 (respondent); if the respondent scored 3 out of 3, they were categorised as 'yes' for having decision-making autonomy and otherwise 'no' [28, 29]. |
| Media exposure | The original questions for this variable were the following: 'Do you listen to the radio at least once a week, less than once a week or not at all?', 'Do you read a newspaper or magazine at least once a week, less than once a week or not at all?' and 'Do you watch television at least once a week, less than once a week or not at all?' [28]. A composite variable was created, combining whether a respondent read newspapers or magazines, listened to radio and/or watched TV: 'no access' if a woman lacked access to all three media; '1' if a woman had access to any of the three media less than once a week; and '2' if a woman had access to any of the three media at least once a week [28, 29]. |
| Antenatal care | The original question asked, 'how many times did you receive antenatal care during the last pregnancy?' [6]. The self-reported response options were the number of visits: 'one visit', 'two visits', 'three visits' and 'four and above visits'. We re-coded this continuous variable as 'no visit', 'at least one visit' and 'four visits' [28, 29]. |
| Place of birth | The original question asked for a self-report of the place where the mother gave birth: 'home', 'public health sector', 'non-government organisation' and 'private health sector'. These were re-coded as 'home delivery' or 'institutional delivery' (encompassing public health sector, non-government organisation and private health sector births) [28, 29]. |
| Number of adverse pregnancy events | Participants were asked if they ever had a pregnancy that miscarried, was aborted or ended in a stillbirth, and if they had ever successfully given birth to a child who later died. A new variable was created as a combination of child mortality (from the birth history question) and pregnancy loss (from the pregnancy history question). To generate this new variable, the number of child deaths from the birth history was first generated. Second, the number of pregnancy losses (miscarriage, abortion or stillbirth) was generated. The status of the index child was excluded from the calculation. Finally, the above two values were added to create the new variable with the following categories: 'no adverse pregnancy events', 'one adverse pregnancy event', 'two adverse pregnancy events' and 'three or more adverse pregnancy events' [28, 29]. |
| Skilled delivery | Participants were asked if they had a skilled health professional—doctor, nurse or midwife, separately—during delivery. This was re-coded as 'yes' if the mother was assisted by one of the above health professionals and 'no' if not [28, 29]. |
| Postnatal check-up | Participants were asked, 'Did anyone check on your health while you were still in the facility within the first 2 days', and the response options were 'yes' or 'no' [28, 29]. |
| Tetanus toxoid injection | The original questions were the following: 'During this pregnancy, how many times did you get a tetanus injection?', 'At any time before this pregnancy, did you receive any tetanus injections?' and 'Before this pregnancy, how many times did you receive a tetanus injection?'. The variable included mothers with two injections during the pregnancy of their last birth or two or more injections (the last within 3 years of the last live birth); three or more injections (the last within 5 years of the last birth); four or more injections (the last within 10 years of the last live birth); or five or more injections at any time prior to the last birth. Categorised as 'no' or 'yes' [28, 29, 32]. |
| Partner age difference | The participants' and their partners' ages were used to calculate the difference in age. The age difference among partners was categorized as: 'husband older by less or equal to five years', 'husband older by five years or more', Wife older by less or equal to five and wife older by five years. However, in this study all husband were older than wife [28, 29]. |
| Sex of the household head | Categorised as 'male' or 'female'. |

(*Continued*)

**Table 1.** (Continued)

| Variable | Category/Measurement/Definition |
|---|---|
| Household wealth index | In the DHS, wealth index was calculated based on household assets, such as televisions and bicycles. Principal components analysis was applied to generate the wealth index as a continuous scale of relative wealth. Wealth index was categorised into five wealth quintiles: 'very poor', 'poor', 'middle', 'rich' and 'very rich'. For this analysis, we re-coded the wealth index as three categories for adequate sampling in each category: as 'poor' (poor and very poor), 'middle' and 'rich' (rich and very rich) [28, 29]. |
| Distance to health facilities | The distance to health facilities was based on participants' subjective ratings. Based on their self-report, the response options were either that the distance to health facilities was a 'big problem' or 'not a big problem'. We used this category as is [28, 29]. |
| **Community-level** | |
| Place of residence | This was recorded as 'rural' or 'urban' in the dataset and was not changed for this analysis [28, 29]. |
| Region of residence | Defined as the region in which the infant's mother was raised. The variable was re-coded as 'agrarian' (encompassing Tigray, Amhara, Oromia, Benishangul, SNNPR, Gambela and Harari regions), 'pastoralist' (Afar and Somali regions) or 'city dweller' (Addis Ababa and Dire Dawa cites). An agrarian society is any community whose economy is based on producing and maintaining crops and farmland. A pastoralist society is any community whose economy is based on raising livestock. A city-dweller society is any city community [28, 29]. |
| Access to improved water | Based on the WHO definition, a household was considered as having access to improved drinking water if it obtained water from sources that protect it from outside contamination, especially fecal matter. An improved source of drinking water comprises one of the following: a piped household connection, public standpipe/borehole, protected dug well or spring, and/or rainwater collection [33]. |
| Access to sanitation | Based on the WHO definition, access to sanitation was the proportion of people with access to either of the following improved sanitation facilities: flush/pour flush toilet connected to the piped sewer system or a septic tank, or toilet [33]. |
| Decision-making autonomy | A combination of three characteristics: decision to obtain healthcare, decision to purchase major household items and decision to visit family. The variable was defined as the proportion of mothers from households that had autonomy make these decisions. Less than 49% was coded as 'poor' and 49% or more was coded as 'good' [28, 29]. |
| Multidimensional Poverty Index | This was defined by multiple deprivations at the household and individual level in three key dimensions: health, education and standard of living. It included 10 indicators: nutrition, child mortality, years of schooling, school attendance, cooking fuel, sanitation, drinking water, electricity, housing and assets. If an individual was deprived of three or more of these, they were categorised as 'poor', and 'not poor' if otherwise [34]. |

*Note.* DHS = Demographic and Health Survey; SNNPR = Southern Nations, Nationalities and Peoples' Region; WHO = World Health Organization.

approximately 10% of the women gave birth when before the age of 20. Nearly half of the households were poor, and the majority of the heads of households were men (see Tables 2–5).

## Determinants of infant mortality

The fixed effects in Model II show the associations between infant mortality and individual- and household-level characteristics when the community-level covariates were not considered, while the fixed effects of Model IV show the associations between infant mortality and individual-household- and community-level determinants. After entering individual-household- and community-level characteristics into Model IV, infant sex was observed to have a significant association with infant mortality; the odds of death before 1 year are approximately 66% higher

**Table 2. Individual-level characteristics of the study population.**

| Individual-level variable | Infant death | | Total |
| --- | --- | --- | --- |
| | Yes (*n* = 495) | No (*n* = 10,528) | |
| Infant sex | | | |
| Male | 325 (65.7) | 5400 (51.3) | 5725 (52.0) |
| Female | 170 (34.3) | 5128 (48.7) | 5298 (48.0) |
| Birth order | | | |
| 1 | 85 (17.2) | 1973 (18.7) | 2058 (18.7) |
| 2–3 | 160 (32.3) | 3200 (30.5) | 3360 (30.5) |
| 4–6 | 137 (27.7) | 3466 (32.9) | 3603 (32.7) |
| $\geq 7$ | 113 (22.8) | 1889 (17.9) | 2002 (18.1) |
| Type of births | | | |
| Singleton | 458 (92.5) | 10,419 (99.0) | 10,877 (98.7) |
| Multiple | 37 (7.5) | 109 (1.0) | 146 (1.3) |
| Reported birth weight | | | |
| Above average | 163 (33.0) | 3323 (32.6) | 3486 (31.6) |
| Average | 172 (34.7) | 4488 (42.6) | 4660 (42.3) |
| Below average | 160 (32.3) | 2717 (25.8) | 2877 (26.1) |
| Preceding birth interval | | | |
| First birth | 93 (18.8) | 1977 (18.8) | 2070 (18.8) |
| < 24 months | 151 (30.5) | 1891 (18.0) | 2042 (18.5) |
| 24–59 months | 193 (39.0) | 5552 (52.7) | 5745 (52.2) |
| $\geq 60$ months | 58 (11.7) | 1208 (11.5) | 1266 (11.5) |
| Maternal age | | | |
| 15–24 years | 114 (23.0) | 2332 (22.2) | 2446 (22.2) |
| 25–34 years | 249 (50.3) | 5593 (53.1) | 5842 (53.0) |
| 35–49 years | 132 (26.7) | 2603 (24.7) | 2735 (24.8) |
| Maternal education | | | |
| No education | 341 (68.9) | 6943 (65.9) | 7284 (66.1) |
| Primary education | 122 (24.6) | 2829 (26.9) | 2951 (26.8) |
| Secondary education | 32 (6.5) | 756 (7.2) | 788 (7.1) |
| Age at first birth | | | |
| < 20 years | 65 (13.1) | 1028 (9.8) | 1093 (9.9) |
| 20–34 years | 348 (70.3) | 7774 (73.8) | 8122 (73.7) |
| 35–49 years | 82 (16.6) | 1726 (16.4) | 1808 (16.4) |
| Total live births | | | |
| 1 | 30 (6.1) | 1405 (13.3) | 1435 (13.0) |
| 2–3 | 171 (34.5) | 3230 (30.7) | 3401 (31.0) |
| 4–5 | 98 (19.8) | 2623 (24.9) | 2721 (24.6) |
| > 5 | 196 (39.6) | 3270 (31.1) | 3466 (31.4) |
| Wealth index | | | |
| Poor | 218 (44.0) | 4939 (46.9) | 5157 (46.8) |
| Middle | 98 (19.8) | 2182 (20.7) | 2280 (20.7) |
| Rich | 179 (36.2) | 3408 (32.4) | 3587 (32.5) |
| Religion | | | |
| Christian | 250 (50.5) | 5955 (56.6) | 6205 (56.3) |
| Muslim | 235 (47.5) | 4326 (41.1) | 4561 (41.4) |
| Other | 10 (2.0) | 247 (2.3) | 257 (2.3) |
| Number of living children | | | |

*(Continued)*

**Table 2.** (Continued)

| Individual-level variable | Infant death | | Total |
|---|---|---|---|
| | **Yes (*n* = 495)** | **No (*n* = 10,528)** | |
| One or less | 147 (29.7) | 1533 (14.6) | 1680 (15.0) |
| 2–3 | 144 (29.1) | 3537 (33.6) | 3681 (33.4) |
| 4–5 | 109 (22.0) | 2824 (26.8) | 2933 (26.6) |
| ≥ 6 | 95 (19.2) | 2634 (25.0) | 2729 (25.0) |
| Number of pregnancy losses (miscarriage, abortion or stillbirth) and child deaths | | | |
| None | 83 (16.8) | 7378 (70.1) | 7461 (67.7) |
| 1 | 248 (50.0) | 2165 (20.6) | 2413 (21.9) |
| 2 | 100 (20.2) | 679 (6.4) | 779 (7.1) |
| ≥ 3 | 64 (12.9) | 306 (2.9) | 370 (3.3) |
| Head of household | | | |
| Male | 441 (89.1) | 9053 (86.0) | 9494 (86.1) |
| Female | 54 (10.9) | 1475 (14.0) | 1529 (13.9) |
| Region | | | |
| Tigray | 21 (4.2) | 695 (6.6) | 716 (6,5) |
| Afar | 7 (1.4) | 108 (1.0) | 115 (1.0) |
| Amhara | 94 (19.0) | 1978 (18.8) | 2072 (18.8) |
| Oromia | 226 (45.7) | 4625 (43.9) | 4851 (44.0) |
| Somali | 29 (5.9) | 479 (4.5) | 508 (4.6) |
| Benishangul | 6 (1.2) | 116 (1.1) | 122 (1.1) |
| SNNPR | 100 (20.2) | 2196 (20.9) | 2296 (20.8) |
| Gambela | 1 (0.2) | 26 (0.2) | 27 (0.2) |
| Harari | 1 (0.2) | 24 (0.2) | 25 (0.2) |
| Addis Ababa | 8 (1.6) | 236 (2.2) | 244 (2.2) |
| Dire Dawa | 2 (0.4) | 45 (0.4) | 47 (0.4) |
| Region, based on way of living | | | |
| Agrarian | 449 (90.7) | 9660 (91.8) | 10,110 (91.7) |
| Pastoralist | 36 (7.3) | 587 (5.6) | 622 (5.6) |
| City dweller | 10 (2.0) | 281 (2.7) | 291 (2.7) |
| Partner education (*n* = 10,462) | | | |
| No education | 225 (47.7) | 4852 (48.6) | 5077 (48.5) |
| Primary education | 195 (41.3) | 3920 (39.2) | 4115 (39.3) |
| Secondary education | 52 (11.0) | 1218 (12.2) | 1270 (12.2) |
| Partner occupation (*n* = 10,462) | | | |
| Not working | 44 (9.3) | 760 (7.6) | 804 (7.7) |
| Professional/business | 38 (8.1) | 1128 (11.3) | 1166 (11.0) |
| Agriculture | 322 (68.2) | 6566 (65.7) | 6888 (65.8) |
| Manual/services | 48 (10.2) | 1162 (11.6) | 1210 (11.6) |
| Other | 20 (4.2) | 375 (3.8) | 295 (3.8) |
| Partner age (*n* = 10,462) | | | |
| 15–24 | 17 (3.6) | 354 (3.5) | 371 (3.5) |
| 25–34 | 209 (44.3) | 3903 (39.1) | 4112 (39.0) |
| 35–64 | 242 (51.3) | 5544 (55.5) | 5786 (55.0) |
| 65+ | 4 (0.8) | 189 (1.9) | 193 (1.8) |
| Age at first marriage | | | |
| < 15 years | 135 (27.3) | 2627 (25.1) | 2762 (25.2) |
| 15–18 years | 177 (35.8) | 3941 (37.6) | 4118 (37.6) |

(*Continued*)

**Table 2.** (Continued)

| Individual-level variable | Infant death | | Total |
|---|---|---|---|
| | Yes (*n* = 495) | No (*n* = 10,528) | |
| > 18 years | 182 (36.9) | 3904 (37.3) | 4086 (37,2) |
| Age difference | | | |
| Woman younger | 8 (1.7) | 235 (2.4) | 243 (2.3) |
| Same age | 14 (3.0) | 176 (1.8) | 191 (1.8) |
| Husband older by ≤ 5 years | 210 (44.5) | 4216 (42.2) | 4426 (42.3) |
| Husband older by ≥ 5 years | 240 (50.8) | 5363 (53.6) | 5363 (53.6) |
| Maternal employment | | | |
| None | 263 (53.1) | 5864 (55.7) | 6127 (55.9) |
| All year | 118 (23.8) | 2161 (20.5) | 2279 (20.8) |
| Seasonal | 89 (18.0) | 1918 (18.2) | 1918 (18.3) |
| Occasional | 25 (5.1) | 585 (5.6) | 585 (5.0) |

*Note*. N = 11,023; values are given as *n* (%). SNNPR = Southern Nations, Nationalities and Peoples' Region.

in male than in female infants, AOR = 1.66, 95% CI [1.25, 2.20], *p* < .001. Infants of multiple births had approximately six times greater odds of dying before the age of 1 month, AOR = 5.8, 95% CI [3.63, 9.37], *p* < .001.

**Table 3. Maternal healthcare service utilisation characteristics for the index child in Ethiopia.**

| Individual-level variable | Infant death | | Total |
|---|---|---|---|
| | Yes (*n* = 495) | No (*n* = 10,528) | |
| Place of delivery | | | |
| Home | 365 (73.7) | 7632 (72.5) | 7997 (72.5) |
| Facility | 130 (26.3) | 2896 (27.5) | 3026 (27.5) |
| Skilled delivery | | | |
| No | 367 (74.1) | 7603 (72.2) | 7970 (72.3) |
| Yes | 128 (25.9) | 2925 (27.8) | 3053 (27.7) |
| Caesarean delivery | | | |
| No | 478 (96.6) | 10,333 (98.1) | 10,811 (98.1) |
| Yes | 17 (3.4) | 195 (1.9) | 212 (1.9) |
| Postnatal check | | | |
| No | 465 (93.9) | 9241 (87.8) | 9706 (88.1) |
| Yes | 30 (6.1) | 1287 (12.2) | 1317 (11.9) |
| Postnatal care within 2 days | | | |
| No | 470 (95.0) | 9710 (92.2) | 10,180 (92.4) |
| Yes | 25 (5.0) | 818 (7.8) | 843 (7.6) |
| Tetanus toxoid injection | | | |
| No | 423 (85.5) | 7603 (72.2) | 8026 (72.8) |
| Yes | 72 (14.5) | 2925 (27.8) | 2997 (27.2) |
| Antenatal care visits (*n* = 7590) | | | |
| None | 124 (25.1) | 2694 (25.6) | 2818 (37.1) |
| ≥ 1 | 64 (13.0) | 2293 (21.8) | 2357 (31.1) |
| 4+ visits | 57 (11.5) | 2357 (22.4) | 2415 (31.8) |

*Note*. N = 11,023; values are given as *n* (%).

**Table 4. Household-level characteristics of the study population.**

| Household-level variable | Infant death | | Total |
|---|---|---|---|
| | Yes (*n* = 495) | No (*n* = 10,528) | |
| Head of household | | | |
| Male | 441 (89.1) | 9053 (96.0) | 9494 (86.1) |
| Female | 54 (10.9) | 1475 (14.0) | 1529 (13.9) |
| Woman's decision-making autonomy | | | |
| No | 207 (42.0) | 3749 (35.6) | 3956 (35.9) |
| Yes | 287 (58.0) | 6779 (64.4) | 7066 (64.1) |
| Access to media | | | |
| No access | 292 (59.0) | 7084 (67.3) | 7376 (66.9) |
| Less than once a week | 102 (20.6) | 1537 (14.6) | 1639 (14.9) |
| At least once a week | 101 (20.4) | 1907 (18.1) | 2008 (18.2) |
| Access to improved water | | | |
| No | 210 (42.4) | 4536 (43.1) | 4746 (43.1) |
| Yes | 285 (57.6) | 5992 (56.9) | 6277 (56.9) |
| Access to improved sanitation | | | |
| No | 457 (92.3) | 9298 (88.3) | 9755 (88.5) |
| Yes | 38 (7.7) | 1230 (11.8) | 1268 (11.5) |

*Note. N* = 11,023; values are given as *n* (%).

In addition to child characteristics that were observed to influence infant survival, certain maternal characteristics were also shown to be associated with child survival within the first year of life. Mothers' health-seeking behaviours, such as prenatal care, were significantly associated with infant mortality. Infants of mothers who received ANC during the last pregnancy were 50% less likely to die in their first year of life compared with infants whose mothers did not receive ANC, AOR = 0.50, 95% CI [0.33, 0.77], *p* = .002 (see Table 6).

In addition to the individual-level characteristics, community-level characteristics (such as the ways of life in the regions in Ethiopia) were significantly associated with infant mortality.

**Table 5. Community-level characteristics of the study population.**

| Community-level variable | Infant death | | Total |
|---|---|---|---|
| | Yes (*n* = 495) | No (*n* = 10,528) | |
| Decision-making autonomy | | | |
| Poor | 258 (52.1) | 5407 (51.4) | 5665 (51.4) |
| Good | 237 (47.9) | 5121 (48.6) | 5358 (48.6) |
| Region, based on way of living | | | |
| Agrarian | 449 (90.7) | 9660 (91.8) | 10,110 (91.7) |
| Pastoralist | 36 (7.3) | 587 (5.6) | 622 (5.6) |
| City dweller | 10 (2.0) | 281 (2.7) | 291 (2.7) |
| Residence | | | |
| Urban | 51 (10.3) | 1165 (11.1) | 1216 (11.0) |
| Rural | 444 (89.7) | 9364 (88.9) | 9808 (89.0) |
| Multidimensional Poverty Index | | | |
| Poor | 12 (2.4) | 305 (3.0) | 317 (2.9) |
| Not poor | 483 (97.6) | 10,223 (97.0) | 10,706 (96.1) |

*Note. N* = 11,023; values are given as *n* (%).

**Table 6. Fixed effects models of infant mortality, using multilevel logistic regression of individual-household- and community-level determinants associated with infant mortality.**

| Variable | Model II | Model III | Model IV |
|---|---|---|---|
| **individual-and household-level variables** | | | |
| Sex | | | |
| Male | 1.68 (1.26, 2.22)* | | 1.66 (1.25, 2.20) |
| Female | 1 (reference) | | 1 (reference) |
| Multiple pregnancy | | | |
| Yes | 8.71 (5.14, 14.75)* | | 8.74 (5.16, 14.83)* |
| No | 1 (reference) | | 1 (reference) |
| Maternal education | | | |
| No education | 0.70 (0.41, 1.23) | | 0.63 (0.35, 1.13) |
| Primary education | 1.03 (0.60, 1.80) | | 0.99 (0.56, 1.76) |
| Secondary or higher | 1 (reference) | | 1 (reference) |
| Number of stillbirths, child deaths | | | |
| None | 1 (reference) | | 1 (reference) |
| 1 | 4.00 (3.00, 5.47)* | | 3.93 (2.90, 5.41)* |
| 2 | 5.30 (3.44, 8.05)* | | 5.20 (3.40, 8.00)* |
| 3 or more | 4.60 (2.64, 7.88)* | | 4.50 (2.62, 7.85)* |
| Postnatal check | | | |
| No | 1 (reference) | | 1 (reference) |
| Yes | 0.99 (0.62, 1.58) | | 0.99 (0.62, 1.60) |
| Tetanus toxoid injection | | | |
| No | 1 (reference) | | 1 (reference) |
| Yes | 0.97 (0.62, 1.58)* | | 0.99 (0.70, 1.41)* |
| Antenatal care visits | | | |
| None | 1 (reference) | | 1 (reference) |
| $\geq 1$ | 0.49 (0.33, 0.72) | | 0.51 (0.34, 0.75) |
| 4+ visits | 0.47 (0.31, 0.72) | | 0.50 (0.33, 0.77) |
| Improved sanitation | | | |
| Yes | 0.87(0.57, 1.33) | | 0.83 (0.52, 1.33) |
| No | 1 (reference) | | |
| Distance to health facility | | | |
| Not a big problem | 1.05 (0.78, 1.69) | | 1.10 (0.82, 1.50) |
| Big problem | 1 (reference) | | 1 (reference) |
| **Community-level variable** | | | |
| Woman's decision-making autonomy | | | |
| High | | 1.01 (0.83, 1.24) | 0.96 (0.73, 1.30) |
| Low | | 1 (reference) | 1 (reference) |
| Region | | | |
| Agrarian | | 1 (reference) | 1 (reference) |
| Pastoralist | | 1.46 (1.16, 1.84)* | 1.44 (1.02, 2.06)* |
| City dweller | | 0.82 (0.48, 1.41) | 0.97 (0.44, 2.15) |
| Residence | | | |
| Urban | | 1 (reference) | 1 (reference) |
| Rural | | 1.87 (1.34, 2.60)* | 1.35 (0.78, 2.33) |
| Multidimensional Poverty Index | | | |
| Poor | | 1 (reference) | 1 (reference) |
| Not poor | | 0.65 (0.43, 1.01) | 1.02 (0.70, 1.50) |

*Note.* Values are given as adjusted odds ratios, with the 95% confidence intervals in parentheses.

**Table 7. Cluster-level random intercept models (measure of variation) of infant mortality, using multilevel logistic regression analysis.**

| Random effect | Model I | Model II | Model III | Model IV |
|---|---|---|---|---|
| Variance (*SE*) | 0.60 | 0.20 | 0.21 | 0.18 |
| ICC (%) | 15.5 | 6.0 | 6.2 | 5.4 |
| PCV (%) | Reference | 61.3 | 60.0 | 65.2 |
| AIC | 4029.46 | 1837.3 | 3947.62 | 1811.92 |

*Note*. 11,023 observations. Model I is the null model, containing no explanatory variable. Model II adjusted for individual-and household-level characteristics; Model III, for community-level characteristics; and Model IV, individual-, household- and community-level characteristics. ICC = intracluster correlation; PCV = proportional change in variance; AIC = Akaike information criterion.

Infants living in pastoralist regions (Somali and Afar) were 1.4 times more likely to die in their first year compared with infants living in agrarian regions (Amhara; Tigray; Oromia; and Southern Nations, Nationalities and Peoples' Region [SNNPR]), AOR = 1.40, 95% CI [1.02, 2.06], *p* = .039 (see Table 6). Furthermore, after adding the individual- household- and community-level characteristics into Model IV, the variation in the odds of infant mortality between communities was statistically significant with $\sigma^2$ = 0.18, *p* = .004. An intraclass correlation coefficient estimated from Model IV indicated that 5.4% of the variability in infant mortality was attributable to differences between community characteristics, and the proportional change in variance from Model I to Model IV was 65.2% (see Table 7).

## Discussion

The rate of infant mortality in Ethiopia is one of the highest in the world. According to the 2016 EDHS, 1 in every 21 children in Ethiopia die before celebrating their first birthday [28]. Several studies have been conducted on the determinants of infant mortality in Ethiopia [8, 10, 21]; however, most have focused on individual determinants only. The present study included individual-, household- and community-level determinants of infant mortality in Ethiopia. The findings from this study indicate that infant sex, multiple pregnancies, number of adverse pregnancy events and ANC are some of the key individual characteristics associated with infant mortality. Beyond these individual characteristics, community characteristics and the region in which an infant lives are also significantly associated with infant mortality.

In this study, infant mortality was higher for boys than for girls; which is the case in most parts of the world [37]. The sex difference was most prominent in the neonatal period [37]. Cross-sectional studies in Ethiopia have reported that male infants are at higher risk of death than female infants; the odds of female infant death are about 20% lower than the odds of male infant death. In another cross-sectional study in Ethiopia, it was shown that the risk of infant mortality was 38% higher among male neonates compared with female neonates [38–41]. This has been explained by sex differences in genetic and biological make-up, with boys being biologically weaker and more susceptible to diseases and premature death [37]. In a study of 75 pooled surveys conducted in 31 countries in SSA, it was reported there were sex differences in mortality among infants [37]. However, other studies have indicated that mortality is higher among girls than boys [42, 43]. The role of harmful traditional practices, such as female genital mutilation, might contribute to the higher female mortality rate; the WHO has reported that female genital mutilation can lead to both immediate and long-term complications [44]. A cross-sectional study of urban Somali on the relationship between female genital mutilation and child mortality indicated that female mortality exceeded male mortality [45]. Another reason for the high rate of infant mortality in females compared with males may be due to sex preference. For families in Asia and Africa, a preferred preference for sons is common [46].

For some families, sons are preferred as they have a higher wage-earning capacity (especially in agrarian economies) and can take care of parents in the later age [47]. For example, in a study from a national survey in India on child gender and parental investment, researchers found that boys received an average of 10% more time and care from their parents than girls did [48]. Another study on Gender and cross-cultural dynamics in Ethiopia also presented that only 20.7% of the study percipients preferred female children [49].

In this study, we found that multiple-birth infants were at higher risk of death compared to singleton births. Multiple births are at high risk for numerous negative birth outcomes, and these outcomes contribute to a higher rate of mortality during the infancy [50, 51]. Different studies have shown that the rate of multiple births in Ethiopia ranges from 14.4 to 37.7 per 1000 deliveries [52–54]. A cross-sectional study in Ethiopia and Zimbabwe showed that multiple births are one of the determinants of infant mortality. According to the 2000 and 2005 health surveys in Ethiopia, multiple births are a serious public health problem [55]. It has been suggested that multiple births increase the economic burden of the family, and this affects the quality of nutrition and health care of the infant [39, 56].

Another finding from this study indicates that ANC service utilisation is significantly associated with infant mortality. As the number of ANC visits increased, the rate of infant mortality decreased, in keeping with past research [57–59]. For example, a systematic review in Ethiopia indicated that the risk of early infant death was lower among women who had four or more ANC visits compared to those who had less than four visits [57].

The findings from this study showed that women (mothers) who had experienced a previous infant loss were twice as likely to experience a subsequent infant death compared to mothers who had no such previous loss. Studies conducted in different developing countries have indicated an effect of previous infant mortality on the survival of the next infant [60]; the death of the previous child may affect the survival of the next child, both biologically and environmentally [60]. The effect of the previous child's death on the survival of the next child may be due to a shorter birth interval [61]. The biological impact of the death of the previous child on the short birth interval operates through an early cessation of breastfeeding and start of ovulation [61]. A shorter birth interval might have a negative impact on the survival status of next birth due to maternal depletion syndrome or the mother not fully recovering from the pregnancy before supporting the next birth [62]. Although the term "maternal depletion syndrome (MDS)" is used to describe the poor health status of mothers, however whether such a syndrome remains unclear [63]. The syndrome was commonly assigned to the nutritional stress induced by successive pregnancies, and pregnancies that were close together [63]. The odds of infant mortality were higher in pastoralist areas (Somali and Afar regions) than in the agrarian areas (Tigray, Amhara, Oromia, SNNPR and Harari regions). The difference in mortality among the regions may be due to variations in service accessibility and coverage. For example, the national Universal Health Coverage (UHC) service capacity and access coverage was 41.1%, 22.0%, 9.5%, 10.6% and 11.7% in Harari, Tigray, Amhara, Oromia and SNNPR, respectively. The Universal Health Coverage (UHC) was lowest in Somali and Afar regions which at 3.7% and 4.1%, respectively [64]. In 2015, the universal health service coverage for Ethiopia was 34.3%, which is substantially behind the SDG target of 80% by the year 2030 but also much higher compared with other African countries [65]. According to a 2015 report, the family planning coverage in the regions of Somali and Tigray was 1.4% and 35.2%, respectively [64]. The immunisation coverage in the regions of Tigray and Afar were also 81.4% and 20.1%, respectively; access to a hospital was 26.1% and 2.3% in the regions of Tigray and Somali, respectively. This huge difference in UHC might lead to variation in population health indicators like infant mortality [64].

The infant health outcomes might not only be due to the differences in family characteristics, but might also be due to differences in the socio-economic characteristics of the

community in which the infant lives [24, 25]. People in a community share similar resources due to most public services being spatially organised into clusters; this determines the health of individuals who live in that community [17]. Evidence suggests that living in an economically and socially deprived community is associated with increased risk of infant mortality. For instance, children born and raised in a community that lacks electricity, improved drinking water and poor access to health facilities are likely to suffer from the same deprivation, which can directly or indirectly influence their health outcomes [14–16]. In a nationally representative study among 5391 live births in Nepal, researchers showed that community factors were associated with infant mortality: infants from the mountain region had a higher rate of mortality compared to those from the lowland region [66]. A possible reason may be people living in mountain areas are particularly vulnerable to food insecurity. Slopes with steep and differing elevations often make the soil shallow, poor in micronutrients, limited, difficult to cultivate and unsuitable for mass agricultural production [67]. In addition, living in mountain areas makes access to health services difficult. For example, a study conducted in rural areas of Ethiopia found that people living in remote areas are at high risk of child mortality [68]. Children who lived 1.5 hours or more hours from a health facility were at a two-fold higher risk of death compared to those who lived within 1.5 hours from a facility [68]. The reason may be that people living in mountainous and remote areas may spend many hours traveling by foot to access maternal and infant health services. In a multicounty study from 28 health surveys, it was shown that being a member of certain families and communities determined child mortality in SSA countries. In the study, household-level characteristics had also a significant effects on infant mortality [17, 18]. In a national study in Malawi, it was found that there was a variation in child mortality at the community level; researchers reported that around 18% of the variation in infant mortality was explained by community-level characteristics [19].

In many areas in Ethiopia, families cannot easily access routine health services. Access to routine health services, and subsequent health outcomes, depends on community-based services and norms. Therefore, intervention at an individual level is insufficient for tackling the problem of infant mortality, because the social and environmental contexts in which an infant lives affects their chances of survival, and people living in the same area share some of the major determinants of infant mortality such as access to water, sanitation and health care. Policymakers could consider community-level interventions, such as improving the community's prevailing norms and attitudes about health behaviours, which can influence the healthcare decisions made by individuals. To minimise infant mortality as a public health problem, the Federal Ministry of Health in Ethiopia should focus on community-based interventions by giving more attention to socially and economically disadvantaged regions.

## Strengths and limitations

While the EDHS is a large-scale, nationally representative dataset, there may be several limitations to this analysis. First, recall bias may be possible, as participants were asked to recall events in the 5 years prior the survey, and they may have forgotten some details. A second limitation of the data is the cross-sectional nature of the survey, which makes it difficult to identify causal relationships between outcome and exposure variables. Third, this study relies on self-reported information from life histories available from a nationally representative survey, which is subject to several sources of error: estimates for specific countries may be affected by these limitations across time, and these need to be taken with caution. Fourth, because of the nature of the survey data, we were unable to make a detailed assessment of the underlying causes of reductions in infant mortality.

While acknowledging these limitations, the EDHS is nationally representative dataset and has been rigorously designed and deployed by the Centres for Disease Control and Prevention using a global framework, and the findings can therefore easily be generalised throughout the country. International comparisons of the findings will also be possible because DHSs adopt similar instruments across countries. We anticipate the findings of this study will have strong policy implications for Ethiopia at the national level, as this is a study that has identified community-level determinants of infant mortality in the country.

## Conclusion

This study showed the importance of individual, household and community determinants in explaining variations in rates of infant mortality in Ethiopia. The results of this study indicate that there may be need to look beyond the influence of individual-level determinants in addressing infant mortality in the country. To ensure a substantial reduction in child mortality during infancy, attention may need to be given to a comprehensive approach comprising community-based interventions aimed at improving child survival in Ethiopia's socially and economically disadvantaged regions. Previous research conducted at the institutional level and at smaller scales limits their applicability to whole nations. The findings from this study may give nationwide insight into the determinants of infant mortality in Ethiopia. Finally, we recommended to policy makers and governments to focus on community level factors in addition to the individual and household level factors to achieve the SDG goals and targets by the end of 2030.

## Acknowledgments

We would like to thank for all women who participated in the Ethiopian Demographic and Health Survey and we would like to thank the DHS Program for allowing us to use the EDHS data for this study.

## Author Contributions

**Conceptualization:** Girmay Tsegay Kiross, Catherine Chojenta.

**Data curation:** Girmay Tsegay Kiross.

**Formal analysis:** Girmay Tsegay Kiross, Daniel Barker, Deborah Loxton.

**Methodology:** Catherine Chojenta, Deborah Loxton.

**Software:** Girmay Tsegay Kiross.

**Supervision:** Catherine Chojenta, Daniel Barker, Deborah Loxton.

**Writing – original draft:** Girmay Tsegay Kiross.

**Writing – review & editing:** Girmay Tsegay Kiross, Catherine Chojenta, Daniel Barker, Deborah Loxton.

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
