## [Decision Letter · Decision Letter 0]

22 Dec 2020

PONE-D-20-29562

Individual and community-level determinants of infant mortality in Ethiopia

PLOS ONE

Dear Dr. Kiross,

Thank you for submitting your manuscript to PLOS ONE. After careful consideration, we feel that it has merit but does not fully meet PLOS ONE’s publication criteria as it currently stands. Therefore, we invite you to submit a revised version of the manuscript that addresses the points raised during the review process.

This is a very interesting paper. The text is clear, but it needs some revision as detailed by the reviewers. Based on the reviewers' comments and my own reading, I believe the paper could improve the contextualization of the problem and provide more detailed on the variables included in the model, please, provide a brief comment explaining what they are exactly capturing, and the expected results.

A few important issues, that could be clarified, are regarding the methodology: a) please, explain in more detailed the exclusion of some community level factors, more detailed on the estimation of ICC and PCV for the logistic models. I would like to see more information on the presentation of each model. It is my understanding that the modeling strategy should first estimate a model that allows  to decompose the variance between the levels of analysis was presented. It does not include explanatory variables, and the error terms act only on the dependent variable, allowing the partition of the total variability found in the data according to the levels of analysis. Based on the null model, more models could be developed, as there is great heterogeneity in the dependent variable. Was that the approach followed? 

Please, see detailed comments by the reviewers and also the attached file. 

We look forward to receiving your revised manuscript.

Kind regards,

Bernardo Lanza Queiroz, Ph.D

Academic Editor

PLOS ONE

Journal Requirements:

"There has been no significant financial support for this work that could have influenced its outcome. "

Reviewers' comments:

Reviewer's Responses to Questions

**Comments to the Author**

1. Is the manuscript technically sound, and do the data support the conclusions?

Reviewer #1: Yes

Reviewer #2: Yes

2. Has the statistical analysis been performed appropriately and rigorously? 

Reviewer #1: Yes

Reviewer #2: I Don't Know

3. Have the authors made all data underlying the findings in their manuscript fully available?

Reviewer #1: Yes

Reviewer #2: Yes

4. Is the manuscript presented in an intelligible fashion and written in standard English?

Reviewer #1: Yes

Reviewer #2: Yes

5. Review Comments to the Author

Reviewer #1: Overall Assessment: The manuscript is well conceived and executed and makes an important contribution to understanding the underlying causes of infant mortality in Ethiopia.

Introduction:

The authors state that the Ethiopian Demographic and Health Survey (EDHS) 2016 has been analyzed for infant mortality before by other researchers. However, they make the case that all earlier studies have examined individual level factors only and did not pay sufficient attention to community level factors. The authors argue, convincingly why other studies failed to take into account shared community level characteristics, which make a significant contribution to high infant mortality rates; they rightly say that these need to be taken into account when considering infant mortality rates, given the objectives attaining SDG 3 goals of reducing infant mortality rates. The literature review section on this aspect is thorough and point to numerous community level factors that should be taken into account while discussing infant mortality in Ethiopia.

Materials and Methods

The method section is straight forward and clearly written, describing the study variables and the community characteristics. While Table 1 appears to be very cumbersome, all the information Is there. The table captures information pertaining to parameters for analysis, including, antenatal care, place of birth, number of adverse pregnancy events, skilled delivery (referring to a doctor, nurse or a midwife). Postnatal checkups, tetanus shots, age of partner, household wealth index, distance from health facility. This table also included the community level factors such as region of residence, quality of water available, sanitation facilities, multidimensional poverty indices. These items were recoded for analysis, recoding of these items.

Results and Data Analysis:

The results (tables 2-7) describe the general characteristics of the study population at the individual and community level. The data analyses consist of descriptive analysis section and a multi-stage multivariate analysis. It requires fairly sophisticated analytical skills. Some of the results are consistent with findings by other investigators. However, they have managed to tease out information that is important.

Suggestions to improve readability.

1. Line 85: - which could be introduces a serious bias - is grammatically wrong. It should read as follows: which could introduce a serious bias;

2. Line 111: the last sentence is a bit confusing need to make sure that the reference is quoted correctly;

3. Line 158: replace -that share with who share;

4. Table 1, page 6: there needs to be an explanation of what century day code means in the context of this study;

5. Table 1 Page 7 reported birth size: It cannot be three fourths (3/4ths) of the births but two thirds (2/3) of the births;

6. Table 1, page 7, reported birth size: delete were from the sentence;

7. Table 1, page 7 reported birth size delete were and write gave birth;

8. Table 1, total live births replace much with many;

9. Table 1 page 7: need to pay attention to tense; since past tense was used earlier, I suggest using was instead of is;

10. Table 1: page 8 age of mother instead of mother's;

11. Table 1, Marital status page 8 marital status and not martial status;

12. Table 1, page 8: marital status - it seems that the categories did not follow what was indicated in the text. Suggest rewriting this section to avoid confusion;

13. Table 1, number of adverse pregnancy events, page 8: replace child with infant in this phrase;

14. Table 1, page 9: should this be the last pregnancy? Please clarify

15. Table 1, place of birth, page 9: this phrase needs to be edited: place where mothers gave not give. the word origination needs to be replaced;

16. Table 1, number of adverse pregnancy events, page 10: replace child with infant in this phrase;

17. table 1, skilled delivery, page 10: insert was in this sentence:

18. Table 1, partner age, page 11: were husbands always older than the wife?

19. Table 1, House-hold wealth, page 11: index the correct spelling; this should be principal and not principle;

20. Table 1, distance from health facility, page 11: was instead of were;

21. Table 1, distance from health facility, page 11: rating not rate;

22. Table 1, distance from the health facility, page 12: add after self report

23. Table 1, distance from health facility, page 12: options not option

24. Table 1, distance from the health facility, page 12: delete it from the sentence to read as is.

25. Table 1, region of residence, Page 12: replace who with that

26. Line 204: had instead of have.

27. Line 210: should be - at an age.

28. Line 214: majority of the head of households were males

29. Line 284: delete a

30. Line 287: The authors have only considered female genital mutilation as a contributing factor. They have not considered the impact of male child preference and neglect of female infants. In India for example, female infant mortality or sex ratio favors male infants. Suggest paper by Prabhat Jha https://arthaimpact.com/wp-content/uploads/2019/09/bc574888-cd46-4f7b-afc2-8d024829057b_35.pdf

31. Line 298: Multiple births have a huge impact on the on the health of the pregnant woman during the perinatal period, including deliveries. I wonder if the authors attempted to correlate C-sections with multiple pregnancies.

32. Line 302: insert a comma after the word increased.

33. Line 303: remove was from this sentence

34. Line 305: rephrase the ....lower among women who attended four and more antenatal care visits

35. Line 308: change sentence compared to women (mothers) who had no such previous loss

36. Line 309: I suggest deleting the sentence Studies ...on the survival of the next infant OR add the references to this statement

37. Line 311: I suggest changing the word survivorship to survival. The meaning of biologically and environmentally is not clear in this context. I suggest removing this altogether or provide more explanation as to why there is an environmental concern.

38. Line 314: insert of before breastfeeding

39. Line 316: I am not familiar with maternal depletion syndrome please include a description.

40. Line 323: do you mean 4.1% instead of 41.%?

41. Line 324 and 325: This sentence needs to be fixed, it is not grammatically correct

42. Line 326: I suggest changing the sentence to - but much higher than- instead of and also much higher compared to other African countries.

43. Line 327-329: these data seem a bit random since not all regions are compared with respect to family planning, immunization coverage and access to hospitals between Tigray, Somali and Afar regions. Could comparative statistics not be located for all the parameters??

44. Line 330: suggest changing the tense to past tense since that is the format used.

45. Line 336-337: this argument seems logical. However, it seems that there are people who belong to the rich category in this group. Was a separate analysis done for the group considered to be rich in infant mortality group?

46. Line 344: the most important reason why infant mortality is high in the mountainous region is the distance from a health post where any assistance can be given to gravid women. Sometimes women have to walk or be carried physically over very difficult terrain to reach the health post. If there is obstructed labor or needing C-section, neither the woman nor the infant survives the journey. Some acknowledgment of this reality should be included in this text.

47. Lines 350-351 This sentence seems to contradict the main argument made in the paper that both community level and individual level factors affect infant mortality. I suggest modifying the statement somewhat.

48. Line 362-365: This could be a stronger statement than the one presented here. Clearly, as suggested in the paper, community level factors include health infrastructure with availability of health facilities and even antenatal services. These come well within the purview of the SDG goals and deserve a stronger recommendation as a conclusion of the analysis.

Reviewer #2: Overall Impression

The manuscript presents an original idea, well executed, and aligned with the scope and broad audience of the journal. Their results support previous evidence of upstream determinants of infant mortality in the Global South and may be impactful and translatable to actions in the country based on. However, I believe that some aspects of this research should be reconsidered. Therefore, I recommend major revisions for this work.

Discussion of specific areas for improvement

The methodological approach implemented in this study seems appropriate for the research question. However, is not very clear the definitions (stated in table 1) and selection of some individual and community- level factors mentioned in the method section that were not part of the main analysis. On one hand, regarding the definition of access to improved water or sanitation at the community level it is not clearly stated whether these variables are capturing the proportion of population with these conditions or whether this is a characteristic of each household included in the study. On the other hand, there are concerns about collinearity among variables at individual and community level that were mentioned in the method section, such as decision-making authority or wealth/ poverty indexes at both individual and community level. As some of these variables were not included in the main analysis, it is important to rule out whether the exclusion correspond to a lack of real association or non-observed effect due to collinearity or incorrect definition of its hierarchical level.

The units of analysis described in the method section correspond to 10,641 children 0-5 years old, while descriptive tables 2-6 are based on (apparently) the number of mothers included in the survey (11,023). I think the authors should clearly state the units of analysis considered and align descriptive tables to this in order to avoid confusion in the readiness of the paper.

Method section needs also to include a brief description of how ICC and PCV measures were calculated for multi-level logistic models as these are different from the way variance component are obtained in linear models.

The study remarks regions as one of the main community-level factors associated to infant mortality. Further in the discussion, the authors explained the potential influence of community-level factors that were not measured in the analysis but considered embedded in the meaning of region (such as access to public services different across regions) on infant mortality. I think that assessing interactions between regions and some individual-level characteristics could bring a more refined approach to the interpretation of these characteristics described in the discussion. If interactions were explored as part of the main analysis and not reported it should be stated in the methods or results section.

Discussion could take leverage of this information to elaborate on the importance of macro- social determinants of infant mortality.

Minor comments

Overall, the manuscript should revise the organization of some sections and tables to improve readability. There are some sentences in the introduction that sounds repetitive (arguments in lines 68-72 are similar to evidence stated in lines 91-94 line 116 similar to 188; 126 similar to 128;) and could be summarized in a more succinct way, which will also allow more room for the description of limitations of previous work in recognizing upstream determinants of infant death and the impact of ignoring this.

Authors should also revise the writing in table 1, being consistent in the way variables are described, and check for missing words/ expression and typos. Tables 2- 6 can be summarized in one table to facilitate the reading. I suggest including in tables 1-6 only variables used in main analysis, to avoid confusion during the reading of the manuscript. Variables considered in previous examination but omitted from main analysis could be referred as supplementary tables.

Edition comments.

Table titles should be reviewed to properly described its contents. Font size should be reviewed throughout tables, and footnote for some marks stated in the tables should be also included

6. PLOS authors have the option to publish the peer review history of their article (what does this mean?). If published, this will include your full peer review and any attached files.

Reviewer #1: No

Reviewer #2: No

---

## [Author Response · Author response to Decision Letter 0]

19 Feb 2021

Reviewer’s comments Author’s response 

1 a) please, explain in more detailed the exclusion of some community level factors, more detailed on the estimation of ICC and PCV for the logistic models. I would like to see more information on the presentation of each model. It is my understanding that the modeling strategy should first estimate a model that allows to decompose the variance between the levels of analysis was presented. It does not include explanatory variables, and the error terms act only on the dependent variable, allowing the partition of the total variability found in the data according to the levels of analysis. Based on the null model, more models could be developed, as there is great heterogeneity in the dependent variable. Was that the approach followed? Thank you very much for these critical questions. We have added the following paragraph: See line 172-174

Backward stepwise multilevel logistic regression analysis was performed to select individual-, household- and community-level variables to each model and those variables with p-value > 0 .25 were removed.

Thank you very much for the critical observation regarding the null model. During model building in this analysis, the null model or model I does not have any contribution to the build the rest of models. In this analysis we started model building with a saturated model i.e. entering all possible variables and removing using a backward stepwise selection. Regarding ICC, it is the measure of variance (random effects), which is the measure of residual errors at individual level and community levels. In this study there was a variation in the odds of infant mortality across communities [1]. 

2 Line 85: - which could be introduces a serious bias - is grammatically wrong. It should read as follows: which could introduce a serious bias; We have made the suggested correction (See line 88).

3 Line 111: the last sentence is a bit confusing need to make sure that the reference is quoted correctly; We have made the following correction (See line 105-107).

In a nationally representative cross-sectional study of 28,647 live births in Nigeria also showed that 16.7% of the variance in the risks of infant mortality across community level characteristics [2]. 

4 Line 158: replace -that share with who share; We have made the appropriate correction and revision (See line 150- 153).

 These are the characteristics of a community or cluster. A community comprises of people living in a particular area or in a common location. In the 2016 Ethiopian Demographic and health survey programmes, the primary sampling units (PSU) are considered as proxies for communities or clusters [3]. 

5 Table 1, page 6: there needs to be an explanation of what century day code means in the context of this study; We have added the following explanation to table 1, page 7).

The century day code is analogous to the century month code and gives the number of days since the beginning of 1900. A century month code (CMC) is the number of the month since the start of the century. For example, January 1900 is CMC 1, January 1901 is CMC 13, and January 1980 is CMC 961[4].

6 Table 1 Page 7 reported birth size: It cannot be three fourths (3/4ths) of the births but two thirds (2/3) of the births; We made correction, two third to three-fourth (See table 1 Page 7).

Measuring the actual weight of the newborn child was very challenging because only one-fourth of Ethiopian mothers gave birth at health facilities; the remaining three-fourth of Ethiopian mothers gave birth at home 

7 Table 1, page 7, reported birth size: delete were from the sentence; We have made the appropriate correction (see table 1, page 7).

8 Table 1, page 7 reported birth size delete were and write gave birth; We have made the appropriate correction (see table 1, page 7).

9 Table 1, total live births replace much with many; We have made the appropriate correction (see table 1, page 8).

10 Table 1 page 7: need to pay attention to tense; since past tense was used earlier, I suggest using was instead of is; We have mad correction (see table 1, page 8).

11 Table 1: page 8 age of mother instead of mother's; We have made the appropriate correction (see table 1, page 8).

12 Table 1, Marital status page 8 marital status and not martial status; We have made the appropriate correction (see table 1, Marital status page 8).

13 Table 1, page 8: marital status - it seems that the categories did not follow what was indicated in the text. Suggest rewriting this section to avoid confusion; We have added the definition of each categories. (see table 1, Marital status page 8).

The original question asked for the marital status of the respondents, and the response options were ‘never married’, ‘married’, ‘living together’, ‘separated’ and ‘widowed’[5]. In this study, ‘never married persons’ are persons who never got married in concordance with valid regulations. “Married persons’ are those who got married before a competent body in concordance with valid regulations. “Widowed persons” are persons whose marriage ceased to exist by death of one of spouses or by declaring a missing spouse dead respectfully. “Divorced persons” are those whose marriage was terminated. “Separated persons” had previously lived with a partner but were not currently living with a partner. ‘living together persons’ are those who living together but have not valid regulations, [6, 7].

14 Table 1, number of adverse pregnancy events, page 8: replace child with infant in this phrase; We have made the appropriate correction (see table 1, page 9).

15 Table 1, page 9: should this be the last pregnancy? Please clarify We have made the appropriate correction (see table 1, page 9). 

We changed the phrase this pregnancy to the last pregnancy.

16 Table 1, place of birth, page 9: this phrase needs to be edited: place where mothers gave not give. the word origination needs to be replaced; We have made the appropriate correction (see table 1, page 9).

17 Table 1, number of adverse pregnancy events, page 10: replace child with infant in this phrase; We have made the appropriate correction (see table 1, page 9).

18 table 1, skilled delivery, page 10: insert was in this sentence: We have made the appropriate correction (see table 1, page 10).

19 Table 1, partner age, page 11: were husbands always older than the wife? Thank you very much for this important question. We have made the following revision. (See table 1, partner age, page 10)

The participants’ and their partners’ ages were used to calculate the difference in age. The age difference among partners was categorized as: ‘husband older by less or equal to five years’, ‘husband older by five years or more’, Wife older by less or equal to five and wife older by five years [6, 7]. However, in this study all husband were older than wife 

20 Table 1, House-hold wealth, page 11: index the correct spelling; this should be principal and not principle; We have made the appropriate correction (see table 1, page 10).

21 Table 1, distance from health facility, page 11: was instead of were; We have made the appropriate correction (see table 1, page 10).

22 Table 1, distance from health facility, page 11: rating not rate; We have made the appropriate correction (see table 1, page 10).

23 Table 1, distance from the health facility, page 12: add after self report We have made the appropriate correction (see table 1, page 11).

24 Table 1, distance from health facility, page 12: options not option We have made the appropriate correction (see table 1, page 11).

25 Table 1, distance from the health facility, page 12: delete it from the sentence to read as is. We have made the appropriate correction (see table 1, page 11).

26 Table 1, region of residence, Page 12: replace who with that We have made the appropriate correction (see table 1, page 11).

27 Line 204: had instead of have. We have made the appropriate correction (see line 188).

28 Line 210: should be - at an age. We have made the appropriate correction (see line 194).

29 Line 214: majority of the head of households were males We have made the appropriate correction (see line 197).

30 Line 284: delete a We have made the appropriate correction (see line 266).`

31 Line 287: The authors have only considered female genital mutilation as a contributing factor. They have not considered the impact of male child preference and neglect of female infants. In India for example, female infant mortality or sex ratio favors male infants. Suggest paper by Prabhat Jha https://arthaimpact.com/wp-content/uploads/2019/09/bc574888-cd46-4f7b-afc2-8d024829057b_35.pdf

Thank you very much for this very important critique and the material

We have added the following paragraph:

Another reason for the high rate of infant mortality in females compared with males may be due to sex preference. For families in Asia and Africa, a preference for sons is common [8]. For some families, sons are preferred as they have a higher wage-earning capacity (especially in agrarian economies) and can take care of parents in later life [9]. For example, in a study from a national survey in India on child gender and parental investment, researchers found that boys received an average of 10% more time and care from their parents than girls did [10]. Another study on Gender and cross-cultural dynamics in Ethiopia also presented that only 20.7% of the study percipients preferred female children [11]. See line 273-281

32 Line 298: Multiple births have a huge impact on the on the health of the pregnant woman during the perinatal period, including deliveries. I wonder if the authors attempted to correlate C-sections with multiple pregnancies. Thank you for this suggestion. In this study C-sections were highly correlated with multiple pregnancies at p-value < 0.001 and for this reason we did not include both variables in the model. We chose to include the multiple birth variable in the model based on the bivariate analysis. 

33 Line 302: insert a comma after the word increased. We have made the appropriate correction (see line 292).

34 Line 303: remove was from this sentence We have made the appropriate correction (see line 293).

35 Line 305: rephrase the ....lower among women who attended four and more antenatal care visits We have rephrased as the following: For example, a systematic review in Ethiopia indicated that the risk of early infant death was lower among women who had four or more ANC visits compared to those who had less than four visits [12]. See line 296-298)

36 Line 308: change sentence compared to women (mothers) who had no such previous loss We have changed the sentence to read: The findings from this study showed that women (mothers) who had experienced a previous infant loss were twice as likely to experience a subsequent infant death compared to mothers who had no such previous loss. (see line 296-298)

37 Line 309: I suggest deleting the sentence Studies ...on the survival of the next infant OR add the references to this statement We have added a reference to this statement (see line 299).

38 Line 311: I suggest changing the word survivorship to survival. The meaning of biologically and environmentally is not clear in this context. I suggest removing this altogether or provide more explanation as to why there is an environmental concern. We have made the appropriate correction (see line 300).

39 Line 314: insert of before breastfeeding We have mad correction (see line 303).

40 Line 316: I am not familiar with maternal depletion syndrome please include a description. We have added the following paragraph (see line 306-310). Although the term “maternal depletion syndrome (MDS)” is used to describe the poor health status of mothers, however whether such a syndrome actually exists remains unclear [13]. The syndrome was commonly assigned to the nutritional stress induced by successive pregnancies, and pregnancies that were close together [13]. 

41 Line 323: do you mean 4.1% instead of 41.%? Thanks, we have mad correction it is 4.1% (see line 317).

42 Line 324 and 325: This sentence needs to be fixed, it is not grammatically correct We have made the following correction (See line 316-317).

Universal Health Coverage (UHC) was lowest in Somali and Afar regions at 3.7% and 4.1%, respectively [14]. 

43 Line 326: I suggest changing the sentence to - but much higher than- instead of and also much higher compared to other African countries. We have made the appropriate correction (see line 323).

44 Line 327-329: these data seem a bit random since not all regions are compared with respect to family planning, immunization coverage and access to hospitals between Tigray, Somali and Afar regions. Could comparative statistics not be located for all the parameters?? We used the findings from the descriptive national health service coverage report, and estimates were not available for all parameters. 

45 Line 330: suggest changing the tense to past tense since that is the format used. We have made the appropriate correction (see line 323).

46 Line 336-337: this argument seems logical. However, it seems that there are people who belong to the rich category in this group. Was a separate analysis done for the group considered to be rich in infant mortality group? We used a wealth index to measure the household income and categorized as poor, middle (average) and rich. Unfortunately the income status of the household is not significantly associated with infant mortality in this study. 

47 Line 344: the most important reason why infant mortality is high in the mountainous region is the distance from a health post where any assistance can be given to gravid women. Sometimes women have to walk or be carried physically over very difficult terrain to reach the health post. If there is obstructed labor or needing C-section, neither the woman nor the infant survives the journey. Some acknowledgment of this reality should be included in this text. Thank you very for this very important comment. 

We have added the following paragraph (See line 340-345).

In addition, living in mountain areas makes access to health services difficult. For example, a study conducted in rural areas of Ethiopia found that people living in remote areas are at high risk of child mortality [15]. Children who lived 1.5 hours or more from a health facility were at a two-fold higher risk of death compared to those who lived within 1.5 hours from a facility [15]. The reason may be that people living in mountainous and remote areas may spend many hours travelling by foot to access maternal and infant health services. 

48 Lines 350-351 This sentence seems to contradict the main argument made in the paper that both community level and individual level factors affect infant mortality. I suggest modifying the statement somewhat. We have mad correction (see line 346-351)

49 Line 362-365: This could be a stronger statement than the one presented here. Clearly, as suggested in the paper, community level factors include health infrastructure with availability of health facilities and even antenatal services. These come well within the purview of the SDG goals and deserve a stronger recommendation as a conclusion of the analysis. Thank you very much. 

We have added the following sentence (see line 390-393).

Finally, we recommended that policy makers and governments focus on community level factors in addition to individual and household level factors to achieve the SDG goals and targets by the end of 2030.

---

## [Editor Report · Decision Letter 1]

1 Mar 2021

Individual-, household- and community-level determinants of infant mortality in Ethiopia

PONE-D-20-29562R1

Dear Dr. Kiross,

We’re pleased to inform you that your manuscript has been judged scientifically suitable for publication and will be formally accepted for publication once it meets all outstanding technical requirements.

Kind regards,

Bernardo Lanza Queiroz, Ph.D

Academic Editor

PLOS ONE
---

## [Editor Report · Acceptance letter]

3 Mar 2021

PONE-D-20-29562R1 

Individual-, household- and community-level determinants of infant mortality in Ethiopia 

Dear Dr. Kiross:

I'm pleased to inform you that your manuscript has been deemed suitable for publication in PLOS ONE. Congratulations! Your manuscript is now with our production department. 

Kind regards, 

on behalf of

Dr. Bernardo Lanza Queiroz 

Academic Editor

PLOS ONE